# ExploreAugment: Adaptive Exploratory Data Augmentation based on Boundary Awareness

## Abstract

Traditional data augmentation often applies uniform transformations across all samples, prioritizing data volume over addressing specific model limitations. This indiscriminate approach can lead to redundant data expansion and inefficient training. We propose ExploreAugment, a novel model-aware data augmentation framework that dynamically targets and refines decision-critical regions in the latent space. Our method first identifies key samples using task-specific selection strategies. Then, it leverages diffusion-based latent interpolation to generate samples that are boundary-ambiguous yet semantically valid. These tailored samples are seamlessly integrated into training via a closed-loop pipeline that continuously adapts to the evolving model state. Extensive experiments across multiple datasets demonstrate that ExploreAugment consistently enhances task performance while significantly reducing augmentation overhead. Notably, our approach outperforms the best baseline by 7.14% on ResNet-50 and 1.75% on DeiT, achieving these gains using only about 15% of the data volume generated by other augmentation methods. This highlights the significant advantage of our boundary-aware, model-driven augmentation for achieving data-efficient learning.[1]

## 1 Introduction

Data augmentation is a fundamental technique in modern machine learning, widely used to expand training datasets, alleviate overfitting, and improve the generalization ability of models (Shorten & Khoshgoftaar, 2019; Cao et al., 2024; Wang et al., 2024a; Chen et al., 2023). Currently, mainstream augmentation methods can be broadly categorized into three types: simple augmentation, mixing-based methods, and distribution-based augmentation. Simple augmentation methods(Krizhevsky et al., 2012; Perez & Wang, 2017), such as flipping, cropping, and rotation, rely on basic transformations to increase data diversity. Mixing methods, like MixUp (Zhang et al., 2018), CutMix (Yun et al., 2019), and their variants (Uddin et al., 2021; Guo et al., 2022; Wu et al., 2023; Shen et al., 2024; Kim et al., 2025; Hu & Wu, 2024), combine existing data points through label mixing or region control, enhancing diversity while maintaining semantic consistency. Distribution-based augmentation (Islam et al., 2024b;a; Tian & Shen, 2025; Zhan et al., 2024; Wang et al., 2024b), using generative approaches such as GAN (Karras et al., 2019) and diffusion models (Rombach et al., 2022), aims to create more realistic samples by altering the data distribution. These methods are typically decoupled from model training and fail to dynamically capture the classifier's weak or uncertain regions. By indiscriminately prioritizing the generation of large volumes of data, they sacrifice sample efficiency, producing an overwhelming number of samples that offer little value to the model's learning process. As a result, there is poor alignment between the generated samples and the training objectives, leading to ineffective learning and excessive computational overhead.

Empirical observations reveal that the introduction of only *small number of highly informative samples* can lead to substantial performance improvements, far exceeding the gains achieved by an equal number of randomly selected samples (Toneva et al., 2019; Paul & Feldman, 2021). This indicates that the key to effective data augmentation is not simply increasing quantity, but rather *targeted enhancement* of the model's most vulnerable regions. Some critical samples exhibit nonlinear gain effects: once properly identified and learned, they can significantly improve the model's understanding of decision boundaries.

---

[1]Code: https://anonymous.4open.science/r/ExploreAugment-3C0C

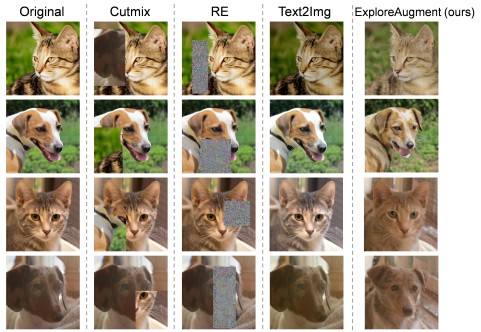

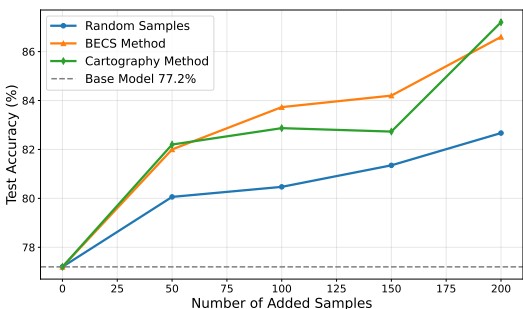

Figure 1: Visual comparison of augmented samples. ExploreAugment generates more coherent and boundary-relevant images than traditional strategies.

Figure 2: Test accuracy gains versus number of added samples, showing BECS and Cartography outperform random sampling, especially with fewer samples.

Motivated by the limitations of static augmentation, we propose a new paradigm—*ExploreAugment*, an exploratory and model-guided data augmentation framework. Unlike conventional methods, ExploreAugment is *dynamic*, *adaptive*, and *structure-aware*: it begins with the current decision state of the model, identifies uncertain or misclassified regions using model-driven indicators such as prediction dynamics and feature similarity (Swayamdipta et al., 2020; Paul & Feldman, 2021), and generates boundary-ambiguous yet semantically coherent samples. These samples are progressively injected into training with incremental weighting, iteratively optimizing the decision boundary. As shown in Figure 1, our method generates samples closer to class boundaries in latent space than traditional augmentations, improving alignment with decision shifts and boosting efficiency in low-data settings.

Our main contributions are summarized as follows:

- We propose a novel exploratory, model-aware data augmentation paradigm that breaks the limitations of traditional static, model-independent approaches.
- We develop a boundary-aware augmentation framework, ExploreAugment, enabling automatic identification of critical regions and targeted sample synthesis,
- ExploreAugment outperforms existing augmentation methods across diverse datasets, improving accuracy over the best baseline by 7.14% and 1.75% under the ResNet-50 and DeiT backbones, using only about 15% of the data volume generated by other augmentation methods.

## 2 RELATED WORK

### 2.1 DATA AUGMENTATION METHODS

Data augmentation is crucial for improving model generalization and can be broadly categorized into simple augmentation, mixing-based methods, and distribution-based augmentation (Shorten & Khoshgoftaar, 2019; Cao et al., 2024; Wang et al., 2024a; Chen et al., 2023). 1) Simple augmentation methods, like flipping, cropping, and rotation (Krizhevsky et al., 2012; Perez & Wang, 2017), use basic transformations to increase data diversity but are static and model-independent, often causing redundancy and inefficiency. 2) Perturbation-based methods such as MultiMix (Shen et al., 2024), GradMix (Kim et al., 2025), PatchMix (Hong & Chen, 2024), and DiffuseMix (Islam et al., 2024a) improve over traditional MixUp/CutMix by incorporating structural or semantic cues. However, they still apply global heuristics and lack adaptivity to model uncertainty, limiting their effectiveness near decision boundaries. 3) Distribution-based augmentation leverages generative models like StyleGAN (Karras et al., 2019), BigGAN (Brock et al., 2019), and Stable Diffusion (Rombach et al., 2022) to synthesize diverse samples. Works such as LatentAugment (Tronchin et al., 2023), Text2Img (He et al., 2022), and REAL-FAKE (Yuan et al., 2024) enhance data quality or diversity via latent perturbation, text conditioning, or real-fake matching. Despite improved sample realism, these methods remain static and model-agnostic, failing to address evolving model weaknesses.

## 2.2 MODEL-GUIDED DATA SELECTION

Model-guided methods identify informative samples using model feedback, including confidence, feature structure, and optimization objectives. Examples include coreset reconstruction (Yang et al., 2024), bi-level optimization (Li et al., 2024), structure-aware selection (Zhang et al., 2024b), and consistency-based filtering for LLMs (Lee et al., 2025). While effective, these methods passively select from existing data and lack the generative capacity to explore new decision-critical regions.[2]

# 3 MOTIVATION

Although extensive research has focused on data augmentation to improve model performance, most existing methods follow the paradigm of "augmentation as transformation", applying indiscriminate strategies that treat all data points equally, without considering the semantic structure of the data or the current state of the model. In fact, instead of insufficient data volume, the key bottleneck often lies in regions with unclear decision boundaries.

Notably, previous studies have shown that a small number of samples close to the decision boundary can significantly improve model performance, outperforming randomly selected samples of equal quantity (Toneva et al., 2019; Paul & Feldman, 2021; Yang et al., 2024; Li et al., 2024; Zhang et al., 2024b). We further validate this by adding equal amounts of boundary samples and random samples to the training set and observing the trend of model performance. We use two selection strategies to identify critical samples:

- **Boundary-aware Entropy and Consistency Selection (BECS):** Selects uncertain and structurally representative samples based on a joint measure of predictive entropy and feature diversity.

- **Dataset Cartography:** Identifies ambiguous samples by analyzing training dynamics, specifically those with high variability (Swayamdipta et al., 2020).

As shown in Figure 2, boundary samples lead to significantly higher accuracy improvements than random sampling, demonstrating that targeted augmentation is more effective than merely increasing data volume.

Motivated by these findings, we propose a novel model-guided data augmentation paradigm. By identifying vulnerable decision regions through model feedback, we synthesize boundary-challenging samples to enhance model performance. Instead of pursuing the expansion of data volume, our methodology shifts to "enhancement of the most vulnerable parts of the model", which improves decision boundaries with less data and lays a groundwork for our model-aware augmentation framework.

# 4 METHOD

We propose **ExploreAugment**, a model-aware data augmentation framework designed to dynamically strengthen classifier decision boundaries through targeted sample generation. As illustrated in Figure 3, the framework follows a closed-loop process that integrates classifier feedback with generative modeling in three stages.

The overall process of ExploreAugment is summarized in Algorithm 1. First, we identify *key samples* near decision boundaries using customized selection strategies based on model uncertainty and latent space structure. Next, we leverage a pretrained diffusion model to synthesize *boundary-ambiguous* yet semantically coherent images by interpolating between selected key samples in the latent space. Finally, the generated samples are selectively injected into the training set, enabling adaptive augmentation and progressive decision boundary refinement.

---

[2]A more detailed discussion of related work is provided in Appendix A.

---

**Algorithm 1** ExploreAugment: Model-Aware Data Augmentation Framework

---

**Input:** Training set $D_{\text{train}}$, classifier $C$, generator $G$ (with encoder $\phi$ and decoder $\psi$), iteration count $T$, key sample count $K$, interpolation factor $\alpha$, selection strategy $S_{\text{select}}$, initial training epochs $E_0$, fine-tuning epochs per round $E_t$

**Output:** Final enhanced classifier $C^*$ trained with adaptive augmentation

1: Train $C$ on $D_{\text{train}}$ for $E_0$ epochs
2: **for** $t = 1$ to $T$ **do**
3:     *// Stage 1: Key Sample Identification*
4:     Evaluate all samples in $D_{\text{train}}$ using classifier $C$
5:     Extract feature embeddings for all samples
6:     Apply the specified strategy $S_{\text{select}}$ (e.g., BECS or Dataset Cartography) to select top-$K$ key samples $S_{\text{key}}$
7:     *// Stage 2: Boundary Sample Generation*
8:     Map $S_{\text{key}}$ to latent space and perform cross-class interpolation
9:     Decode interpolated representations to obtain generated sample set $X_{\text{gen}}$
10:     *// Stage 3: Adversarial Fusion Training*
11:     Filter $X_{\text{gen}}$ using classifier $C$
12:     Assign training weights to selected samples
13:     Augment training set: $D_{\text{aug}} \leftarrow D_{\text{train}} \cup X_{\text{gen}}$
14:     Fine-tune $C$ on $D_{\text{aug}}$ for $E_t$ epochs
15:     $D_{\text{train}} \leftarrow D_{\text{aug}}$
16: **end for**
17: **return** $C^*$

---

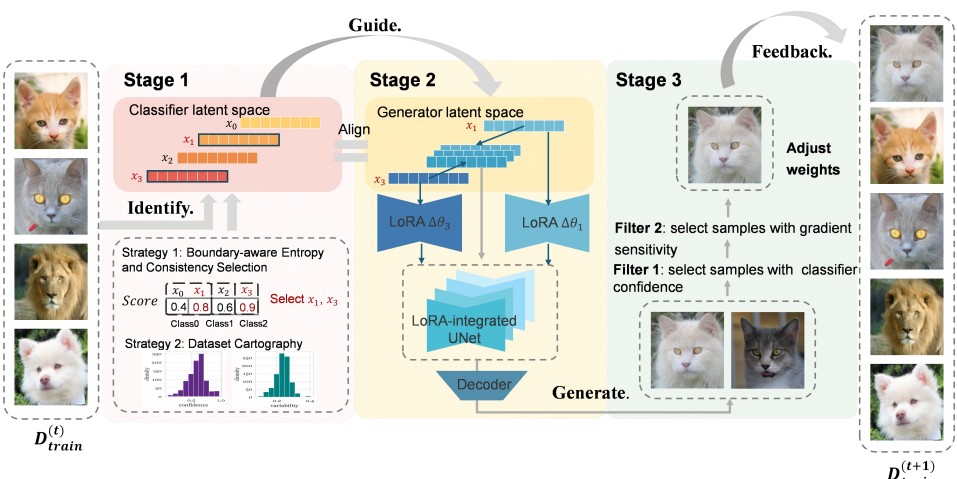

Figure 3: Overall process of ExploreAugment. Stage 1 identifies critical samples under the current model state based on specific selection strategies. Stage 2 maps the selected samples into the latent space of a generative model under the guidance of the classifier, and performs interpolation between samples from different classes to synthesize new samples near the decision boundaries. Stage 3 filters and reweights the generated data before adding them back for further training.

## 4.1 STAGE 1: KEY SAMPLE IDENTIFICATION

In classification, boundary samples often show high uncertainty and lie far from class centers in latent space. We employ two flexible strategies to identify such informative regions for targeted augmentation.

**(1) BECS.** Samples near classification boundaries often have high uncertainty and are far from class prototypes. Relying on a single metric is insufficient: entropy may capture noise, while cosine similarity may overlook ambiguous samples. Therefore, we propose a composite scoring function that combines both metrics for a growing subset $\mathcal{S}$ of selected samples. The score of a sample $x_i$ is defined as:

$$\text{Score}(x_i) = (1 - \alpha) \cdot \text{sim}(x_i) + \alpha \cdot H_i, \tag{1}$$

where

$$\text{sim}(x_i) = \frac{1}{|\mathcal{S}|} \sum_{x_j \in \mathcal{S}} \frac{f_i \cdot f_j}{\|f_i\| \|f_j\|}, \tag{2}$$

and $\alpha \in [0, 1]$ is a trade-off coefficient between entropy $H_i$ and structural similarity. A greedy selection process [3] is applied to select the top-$k$ samples per class with the highest composite scores.

**(2) Dataset Cartography.** Following Dataset Cartography (Swayamdipta et al., 2020), we track per-sample prediction dynamics across epochs to identify unstable or uncertain samples. For each training sample $x_i$, let $p^{(e)}(y_i^* | x_i)$ be the model's predicted probability for the ground truth class in epoch $e$. We compute two metrics:

$$\text{Confidence}(x_i) = \frac{1}{E} \sum_{e=1}^{E} p^{(e)}(y_i^* \mid x_i), \tag{3}$$

$$\text{Variability}(x_i) = \text{std} \left( \{ p^{(e)}(y_i^* \mid x_i) \}_{e=1}^{E} \right). \tag{4}$$

Samples with high variability and low confidence are labeled as *ambiguous* and typically lie near class boundaries. These samples serve as valuable guides for generation, enabling the classifier to focus on decision-critical regions. [4]

The two strategies offer alternative approaches for identifying informative samples and can be flexibly adopted based on the characteristics and requirements of the target task. Experiments show that for larger models such as ResNet-50, Dataset Cartography yields better results, while for smaller models like ResNet-34, BECS performs more effectively.

### 4.2 STAGE 2: BOUNDARY SAMPLE GENERATION

Once key samples are identified, high-quality boundary images are generated via diffusion-based interpolation. The process consists of three parts:

**Latent Space Mapping.** Given selected key samples from the downstream classifier, we map their features from the classifier latent space $\mathcal{Z}_c$ to the latent space of the pretrained Stable Diffusion model $\mathcal{Z}_s$. We train a mapping function $f : \mathcal{Z}_c \to \mathcal{Z}_s$[5], ensuring the generator can decode classifier-aware representations into semantically meaningful images.

**Latent Interpolation.** For a selected pair $(x_a, x_b)$ from different classes, we follow the latent interpolation strategy proposed in DiffMorpher (Zhang et al., 2024a). Specifically, we first fine-tune the diffusion model using LoRA to obtain sample-specific low-rank updates $\Delta\theta_a$ and $\Delta\theta_b$ for the UNet parameters $\theta$. Each LoRA $\Delta\theta_i$ is trained to minimize the denoising loss on the VAE latent embedding $z_{0i}$ of the input image $x_i$:

$$\mathcal{L}(\Delta\theta_i) = \mathbb{E}_{\epsilon, t} \left[ \left\| \epsilon - \epsilon_{\theta + \Delta\theta_i} \left( \sqrt{\bar{\alpha}_t} z_{0i} + \sqrt{1 - \bar{\alpha}_t} \epsilon, t, c_i \right) \right\|^2 \right], \quad i \in \{a, b\}, \tag{5}$$

where $\epsilon \sim \mathcal{N}(0, I)$ is Gaussian noise, $c_i$ is the text embedding, and $\epsilon_{\theta + \Delta\theta_i}$ denotes the LoRA-integrated UNet. Each $\Delta\theta_i$ is optimized separately via gradient descent. Then compute a spherical linear interpolation (Slerp) in the latent space:

$$\hat{x}_\lambda = \text{Slerp}(x_a, x_b; \lambda), \quad \hat{\theta}_\lambda = (1 - \lambda) \cdot \theta_a + \lambda \cdot \theta_b. \tag{6}$$

The pair $(\hat{x}_\lambda, \hat{\theta}_\lambda)$ is fed into the diffusion model to synthesize ambiguous samples near the semantic boundary. Adjusting $\lambda \in [0, 1]$ allows control over class similarity.

---

[3]The detailed algorithm can be found in Appendix B.

[4]The datamap of AFHQ is provided in Appendix C.

[5]Details of the mapping function $f$ and its training procedure are provided in Appendix D.

**Label Assignment.** Considering that generated images are obtained via interpolation in a latent space specifically adapted through LoRA fine-tuning on the endpoint images, relying on classifier outputs for pseudo-labeling can be unreliable. Instead, we assign hard labels based on proximity in this adapted latent space: for a generated sample $\hat{x}_\lambda$, we determine its nearest neighbor among the fine-tuned endpoint images, and assign the corresponding true label (e.g., $y_a$ if closer to $x_a$). Since the latent space is specifically optimized to capture the high-level semantics of these endpoint images, nearest-neighbor assignment provides a robust and accurate labeling strategy.

### 4.3 STAGE 3: ADVERSARIAL FUSION TRAINING

To enable joint evolution of the classifier and generator, we introduce a progressive training scheme that injects generated samples based on model state.

**Dynamic Sample Injection.** Only real samples are used in the initial stage of training. After several epochs, key regions are identified, and new synthetic samples are added. We adjust their contribution via a scheduling function $w(t)$, implemented as a cosine ramp-up:

$$
w(t) = \begin{cases} 0, & t < t_0, \\ \frac{1}{2}\left(1 - \cos\left(\pi \frac{t-t_0}{T_{\text{ramp}}}\right)\right), & t \in [t_0, t_0 + T_{\text{ramp}}], \\ 1, & t > t_0 + T_{\text{ramp}}. \end{cases}
\tag{7}
$$

where $t_0$ is the epoch at which synthetic sample injection begins, and $T_{\text{ramp}}$ controls the duration of the ramp-up. The total training loss is defined as:

$$
\mathcal{L}_{\text{total}} = \mathcal{L}_{\text{ce}}(\mathcal{D}_{\text{real}}) + w(t) \cdot \mathcal{L}_{\text{ce}}(\mathcal{D}_{\text{syn}}),
\tag{8}
$$

ensuring that synthetic samples gradually influence learning, in alignment with the model's uncertainty and its evolving decision boundaries.

**Auxiliary Selection Mechanism.** To ensure the quality of synthetic samples, we introduce a two-stage filter. First, select samples with classifier confidence in $[p_{\min}, p_{\max}]$:

$$
\hat{\mathcal{X}}_1 = \{\hat{x}_i \in \hat{\mathcal{X}} \mid p_{\min} \leq C^{(t)}(\hat{x}_i) \leq p_{\max}\},
\tag{9}
$$

where $C^{(t)}(\hat{x}_i)$ denotes the confidence score given by the classifier at training epoch $t$ for the sample $\hat{x}_i$. Note that this confidence is different from the previously defined equation 3, as $C^{(t)}(\hat{x}_i)$ reflects the confidence at a specific training epoch, while equation 3 measures the averaged confidence over multiple epochs. Then compute gradient sensitivity:

$$
s_i = \left\| \frac{\partial C^{(t)}(\hat{x}_i)}{\partial \hat{x}_i} \right\|_2.
\tag{10}
$$

Select the top-$k\%$ most sensitive samples:

$$
\hat{\mathcal{X}}' = \text{Top-}k\% \left(\hat{\mathcal{X}}_1, s_i\right).
\tag{11}
$$

Finally, merge the selected set with the real data to form the training set for the next iteration:

$$
\mathcal{D}_{\text{train}}^{(iter+1)} = \mathcal{D}_{\text{train}}^{(iter)} \cup \hat{\mathcal{X}}',
\tag{12}
$$

This closed-loop feedback continuously adapts to the classifier's decision bottlenecks, enabling generator-classifier co-evolution. Unlike static pre-processing, generation becomes a dynamic part of training.

In summary, ExploreAugment enables dynamic coordination between classifier and augmentation through key sample discovery, boundary-aware generation, and progressive training. This approach improves coverage of decision-critical regions, and remains flexible and model-adaptive.

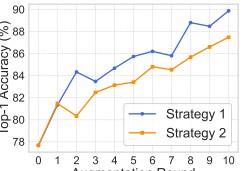 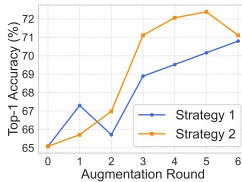 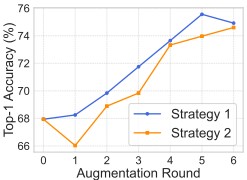

(a) ResNet-50 on AFHQ    (b) ResNet-34 on AFHQ    (c) ResNet-50 on Flowers    (d) ResNet-34 on Flowers

Figure 4: Accuracy comparison under two sample selection strategies across different datasets and model capacities. Strategy 2 performs better on the larger model (ResNet-50), while Strategy 1 shows advantages on the smaller model (ResNet-34), suggesting that strategy design should align with model capacity.

Table 1: Top-1 accuracy and augmentation ratio (i.e., number of added samples relative to original data) of different augmentation methods across datasets.

| Method | AFHQ | Flowers | Birds | Pets | Aug. Ratio |
|---|---|---|---|---|---|
| **ResNet-50 backbone** | | | | | |
| Only real data | 91.07 | 65.08 | 86.97 | 58.21 | – |
| Cutout | 93.00 | 68.57 | 90.02 | 65.72 | 1× |
| Random Erase | 90.60 | 69.52 | 91.05 | 63.05 | 1× |
| Text2Img | 91.47 | 70.16 | 88.57 | 66.20 | 0.6-1× |
| YONA | 93.47 | 71.11 | 88.60 | 65.81 | 1× |
| DiffMix | 92.73 | 66.10 | 93.73 | 65.84 | 0.6-1× |
| Ours (BECS) | 95.73 | 70.79 | **94.13** | 68.97 | **0.1-0.2×** |
| Ours (Cartography) | **96.78** | **72.38** | 93.82 | **73.34** | **0.1-0.2×** |
| **DeiT backbone** | | | | | |
| Only real data | 95.27 | 95.87 | 92.80 | 87.67 | – |
| Cutout | 98.47 | 96.51 | 93.64 | 89.07 | 1× |
| Random Erase | 98.07 | 97.78 | 93.03 | 88.62 | 1× |
| Text2Img | 99.20 | 91.11 | 93.68 | 88.80 | 0.6-1× |
| YONA | 99.13 | 96.51 | 90.01 | 89.12 | 1× |
| DiffMix | 99.40 | 94.92 | 92.84 | 88.55 | 0.6-1× |
| Ours (BECS) | **99.67** | 97.46 | **95.43** | 89.89 | **0.1-0.2×** |
| Ours (Cartography) | 99.53 | **98.10** | 94.56 | **89.98** | **0.1-0.2×** |

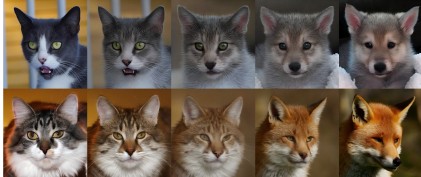

(a) DiffMorpher generates semantically coherent latent space transitions.

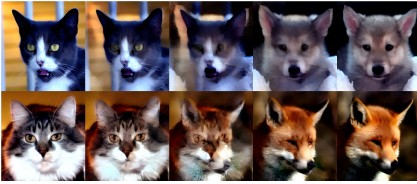

(b) VAE interpolation resembles a simple image blend.

Figure 5: Comparison of image interpolation methods.

## 5 EXPERIMENT

### 5.1 EXPERIMENTAL SETTINGS

**Datasets.** We select diverse, fine-grained image classification datasets, including AFHQ (Choi et al., 2020), Birds-525-Species (Kotha, 2023), Flowers (Nilsback & Zisserman, 2008), and Oxford Pets (Parkhi et al., 2012). These datasets collectively cover various domains and category granularity, validating the universality and stability of our method.

**Model Architectures.** We employ two representative backbones: ResNet (He et al., 2016) as a classic CNN, and DeiT (Touvron et al., 2021) as a lightweight vision transformer, which allow us to assess the adaptability of ExploreAugment across different model inductive biases.

### 5.2 MAIN RESULTS

Table 1 presents the Top-1 accuracy of various data augmentation methods across four benchmark datasets using both ResNet-50 and DeiT backbones. Compared to conventional and recent augmentation techniques such as Cutout, Random Erasing, Text2Img, YONA, and DiffMix, our proposed ExploreAugment consistently achieves superior performance across all settings.

Under the ResNet-50 backbone, ExploreAugment yields the highest accuracy on all datasets. Specifically, the BECS strategy improves upon the only real data baseline by up to +7.3% on the Flowers dataset and +7.16% on Birds, demonstrating its effectiveness in addressing fine-grained recognition tasks. Moreover, the Cartography-based variant further enhances performance, achieving 96.78% on AFHQ and 73.34% on Pets, highlighting its capability to identify and enhance decision-critical regions. These improvements are particularly significant on smaller or more challenging datasets, where traditional augmentation methods offer limited gains.

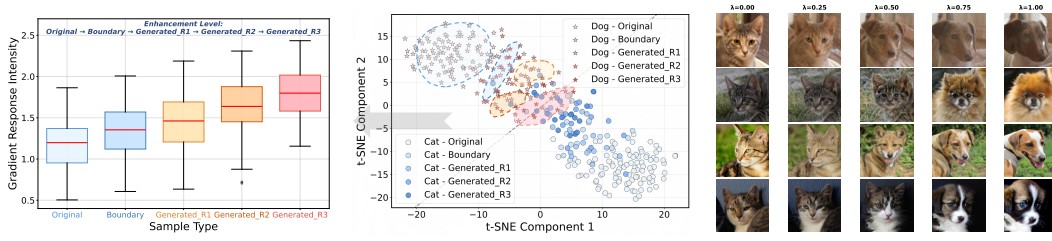

(a) Gradient sensitivity analysis     (b) Latent space distribution     (c) Generate sample sequence

Figure 6: Visualizations of ExploreAugment. (a) Gradient response magnitudes for original, boundary, and generated samples at different interpolation levels. (b) t-SNE visualization of sample groups in latent space. (c) Semantic interpolation enables smooth transitions between classes.

Even with the DeiT backbone, which already benefits from global self-attention mechanisms, ExploreAugment still provides substantial improvements. BECS achieves 99.67% on AFHQ and 95.43% on Birds, surpassing all baselines. Cartography reaches the highest accuracy of 98.10% on Flowers and 89.98% on Pets. These results show that even for transformer-based models, targeted boundary-aware augmentation can yield additional performance gains.

Importantly, these improvements are achieved by adding only 10% to 20% new samples relative to the original dataset size, significantly less than other methods such as Cutout or Text2Img, which often require augmenting by 60% to 100% or more.[6] This highlights the method's efficiency in delivering targeted performance gains with minimal data increase.

Overall, ExploreAugment outperforms both conventional and generative augmentation methods. It leads to more informative samples, improved efficiency, and stronger model performance, highlighting the benefits of dynamic, model-aware augmentation over static, quantity-driven approaches.

## 5.3 ABLATION STUDY

**Comparison of Sample Selection Strategies.** We compare two sample selection strategies under equal training epochs, as shown in Figure 4. On the subset of AFHQ and Flowers datasets, Strategy 1 slightly outperforms Strategy 2 when using the ResNet-34 backbone. However, Strategy 2 achieves better performance on ResNet-50. These results indicate that model capacity influences the preference for sample selection, suggesting that strategy design should align with the model's representational capability.

**Comparison of Image Generation Methods.** Figure 5 shows samples generated by DiffMorpher (Zhang et al., 2024a), which produces smooth transitions between two input images and yields boundary samples that naturally combine the semantic and visual traits of both endpoints. Compared with a VAE-based linear interpolation baseline, which often results in simple blends lacking structural continuity and semantic consistency, DiffMorpher better captures continuous latent trajectories and delivers superior visual quality and semantic coherence.

## 5.4 BOUNDARY UNDERSTANDING ANALYSIS

To better understand how our augmentation framework targets decision-critical regions, we conduct a boundary analysis through gradient response visualization and latent space projection. Figure 6a shows gradient responses for original, boundary, and generated samples across three augmentation rounds. Boundary and generated samples exhibit stronger gradients than originals, indicating higher sensitivity to the classifier's decision function. Figure 6b visualizes their t-SNE projections, revealing a gradual shift toward the decision boundary and increasing semantic ambiguity. Figure 6c illustrates interpolation trajectories where generated samples progressively approach the boundary. These results confirm that ExploreAugment produces boundary-aware samples that refine the model's decision boundary.

---

[6]Detailed augmented sample configurations are provided in Appendix E.

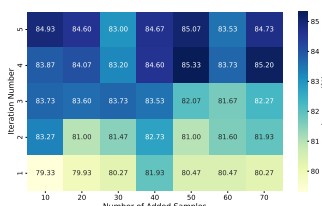 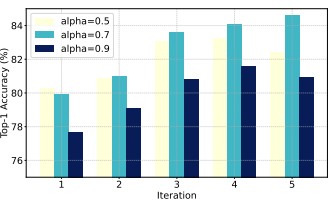 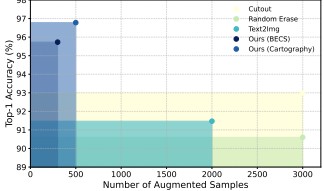

(a) Accuracy heatmap across rounds.  (b) Accuracy under different $\alpha$.  (c) Sample efficiency comparison.

Figure 7: Performance analysis under different hyperparameters and augmentation strategies.

## 5.5 HYPERPARAMETER ANALYSIS

This section analyzes three key hyperparameters of ExploreAugment, the added samples per iteration, iteration count, and boundary control parameter $\alpha$, to guide practical parameter tuning.

**Impact of Sample Quantity and Iteration Count.** To evaluate ExploreAugment under different settings, we tested how the number of generated samples per round and total iterations affect classification accuracy on an AFHQ subset using ResNet-50 (He et al., 2016). As shown in Figure 7a, accuracy steadily improves with more iterations (e.g., adding 10 samples per round raises accuracy from 79.33% to 84.93%), but exceeding 50 samples per round slightly degrades performance due to noise or redundancy. We recommend 4–5 rounds with each round adding about 3–5% of the original training size to balance performance and efficiency.

**Effect of Boundary Control Parameter $\alpha$.** We further analyze the impact of the boundary control parameter $\alpha$ shown in equation 1, which regulates how close the generated samples are to the classifier decision boundary in latent space. We test $\alpha \in \{0.5, 0.7, 0.9\}$ across 1 to 5 rounds of augmentation, as shown in Figure 7b. $\alpha = 0.7$ yields the best accuracy with steady improvement, while $\alpha = 0.5$ is stable but slightly lower, and $\alpha = 0.9$ performs worst, showing that overly strict boundary constraints hinder latent exploration. We recommend $\alpha = 0.7$ to balance boundary alignment and generative diversity.

## 5.6 SAMPLE EFFICIENCY EVALUATION

To assess the efficiency of different augmentation strategies, we compare classification performance with respect to the number of added samples. As shown in Figure 7c, our method achieves superior performance using significantly fewer samples compared to conventional approaches.

On the subset of the AFHQ dataset with the ResNet-50 backbone, BECS and Cartography achieve accuracies of 95.73% and 96.78%, outperforming Cutout at 93.00%. Remarkably, our methods use only 10% and 16.7% of the sample volume required by Cutout, yet deliver over 2.7 and 3.7 percentage points of improvement, respectively. Similarly, compared to Text2Img, which attains 91.47%, our methods achieve significantly higher accuracy with just 15% of the sample size.

These results show that samples generated by ExploreAugment are substantially more informative than those from traditional or generic generative methods. Notably, other methods fail to improve performance even when generating far more images, indicating a clear performance ceiling. Our approach requires only a modest increase in generation time but delivers meaningful accuracy gains by focusing on decision-critical regions near the classifier's boundary[7].

## 6 CONCLUSION

We introduced ExploreAugment, a model-aware data augmentation framework that improves classifier performance by focusing on decision boundary regions. By combining key sample identification, diffusion-based boundary sample generation, and adversarial fusion training, ExploreAugment generates semantically coherent and informative samples that challenge the classifier. This closed-loop design adapts augmentation based on model uncertainty, refining decision boundaries. Experiments demonstrate the effectiveness of our approach in balancing sample quality and quantity. Future work includes extending ExploreAugment to multimodal latent spaces and integrating it with active learning to optimize sample generation and selection.

---

[7]Further experimental details are provided in Appendix F.

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

APPENDIX

## A  RELATED WORK

### DATA AUGMENTATION METHODS

Data augmentation plays a critical role in improving model generalization and robustness by artificially enlarging the training set and exposing models to richer variations of the data distribution. Existing approaches can be broadly categorized into *simple augmentation*, *mixing-based perturbation*, and *distribution-based generation* (Shorten & Khoshgoftaar, 2019; Cao et al., 2024; Wang et al., 2024a; Chen et al., 2023), each of which focuses on different aspects of sample diversity. Table 2 summarizes the representative methods and their key characteristics.

**Simple augmentation methods** apply basic transformations to increase data diversity but remain static and model-independent. Typical operations such as flipping, cropping, rotation, and color jittering (Krizhevsky et al., 2012; Perez & Wang, 2017) enrich low-level variations and reduce overfitting, thereby improving baseline accuracy in many vision tasks. However, these transformations merely manipulate superficial appearance without exploiting semantic relationships among samples or adapting to the model's current decision state. Consequently, they often lead to redundant samples and provide limited performance gains once the model has been exposed to a sufficiently varied dataset.

**Perturbation-based augmentation** enhances data diversity by creating synthetic samples through controlled perturbations of existing data. Recent techniques extend classic mixup strategies by incorporating structural priors or adaptive selection. For example, MultiMix (Shen et al., 2024) generates multiple interpolations per sample pair to stabilize optimization and enrich intermediate representations, while GradMix (Kim et al., 2025) leverages gradient-based selection to perform adaptive MixUp that is particularly effective in class-incremental learning scenarios. PatchMix (Hong & Chen, 2024) operates at the patch level, mixing local regions across images to simultaneously enhance local feature robustness and preserve global semantics. Another representative approach, DiffuseMix (Islam et al., 2024a), integrates diffusion models to refine mixup outputs and enforce structural consistency in the synthesized images. Although these methods substantially improve on earlier mix-based strategies through structural or semantic guidance, they generally remain unaware of the downstream model's evolving decision boundary and thus fail to specifically target semantically ambiguous or decision-critical regions during training.

**Distribution-based augmentation** has received increasing attention in recent years with the rapid advancement of generative models. State-of-the-art generators such as StyleGAN (Karras et al., 2019), BigGAN (Brock et al., 2019), and Stable Diffusion (Rombach et al., 2022) can synthesize high-fidelity and diverse images that closely match the distribution of real data, enabling effective dataset expansion and improved performance in long-tailed or low-resource scenarios. Building on these capabilities, LatentAugment (Tronchin et al., 2023) perturbs latent representations in pretrained generative models to produce high-value samples near class boundaries, where discriminative information is most critical. Text2Img (He et al., 2022) explores the use of text-to-image generative models to create labeled training data and systematically evaluates the impact of prompt-driven synthesis on image recognition tasks. REAL-FAKE (Yuan et al., 2024) further improves distributional relevance by constructing real–fake pairs that closely match the statistics of true data, thereby enhancing both sample diversity and model relevance. Despite their impressive ability to generate visually diverse and realistic samples, most generative augmentation methods still focus on global data diversity rather than being guided by the model's current decision weaknesses, limiting their ability to target the most informative or uncertain regions of the input space.

### MODEL-GUIDED DATA SELECTION

Complementary to augmentation, a growing body of research explores leveraging model signals and training dynamics to guide the selection of valuable samples. Such approaches aim to identify data points that are most informative for improving model performance or refining decision boundaries, thereby reducing training cost without sacrificing accuracy. For example, Boundary-Set(Yang et al., 2024) propose a coreset selection method that explicitly reconstructs the decision boundary, achieving over 50% data reduction on ImageNet-1K with minimal accuracy loss. Boundary matters(Li et al., 2024) design a bi-level optimization framework that jointly balances sam-

Table 2: Comparison of representative methods. ✓ denotes full support, (✓) denotes partial support, and — denotes no support.

| Method | Model-Aware | Generative | Boundary-Targeted | Adaptive | Sample Efficiency |
|---|---|---|---|---|---|
| Flipping/Cropping | — | — | — | — | — |
| MultiMix | (✓) | — | — | (✓) | (✓) |
| GradMix | ✓ | — | (✓) | ✓ | (✓) |
| PatchMix | (✓) | — | — | (✓) | (✓) |
| DiffuseMix | (✓) | ✓ | — | (✓) | (✓) |
| StyleGAN | — | ✓ | — | — | — |
| BigGAN | — | ✓ | — | — | — |
| Stable Diffusion | — | ✓ | — | — | — |
| LatentAugment | — | ✓ | (✓) | — | (✓) |
| Text2Img | — | ✓ | — | — | — |
| REAL-FAKE | — | ✓ | (✓) | — | (✓) |
| BoundarySet | ✓ | — | ✓ | ✓ | ✓ |
| Boundary matters | ✓ | — | ✓ | ✓ | ✓ |
| CDS | ✓ | — | ✓ | ✓ | ✓ |
| Dataset Cartog. | ✓ | — | (✓) | ✓ | ✓ |
| **ExploreAugment (ours)** | ✓ | ✓ | ✓ | ✓ | ✓ |

ple diversity and boundary uncertainty, effectively selecting informative samples for fine-tuning in resource-constrained settings. The "Contributing Dimension Structure" (CDS) approach (Zhang et al., 2024b) measures the structural importance of deep feature dimensions to improve coreset representativeness and maintain class balance. In addition, recent work extends dataset cartography to large-scale language data, where mean–variance training dynamics are used to identify high-quality alignment samples for large language models (Lee et al., 2025). These methods demonstrate that model-informed selection can substantially enhance training efficiency and performance by focusing learning on decision-critical data.

However, while these strategies are highly effective for identifying boundary or ambiguous examples, they operate passively over existing data and lack the ability to actively synthesize informative samples in unexplored semantic regions. **ExploreAugment** addresses this gap by combining key sample selection with generative modeling to create cross-boundary samples and establish a closed-loop training system that co-evolves the model and its training data, thereby unifying the strengths of targeted selection and data generation.

## B    BECS ALGORITHM

As shown in Algorithm 2, during the sample selection process, for each class $c$, we aim to identify $k$ representative boundary-neighborhood samples. Initially, a single seed sample is randomly selected from all samples of class $c$ and added to the selected set $\mathcal{S}_{\text{selected}}$. Then, the algorithm iteratively selects additional samples from the remaining candidates in class $c$, based on a combined scoring function.

This scoring function considers two factors:

- **Mean cosine similarity:** Measures the average similarity between a candidate sample and all samples already in the selected set, capturing how different the candidate is from already selected ones.

- **Predictive entropy:** Reflects the uncertainty of the candidate and its proximity to the decision boundary.

By jointly considering diversity and uncertainty, the sample with the highest combined score is added to the set in each iteration, until $k$ samples are selected. This process is repeated independently for all classes, resulting in a final set of selected samples $\mathcal{S}$ used for downstream augmentation or training.

---

**Algorithm 2** Boundary-aware Entropy and Consistency Selection

---

    **Input:** Feature set $F = \{f_1, \ldots, f_n\}$, entropy values $H = \{H_1, \ldots, H_n\}$, labels $Y = \{y_1, \ldots, y_n\}$, number of classes $C$, per-class sample count $k$, trade-off coefficient $\alpha$
    **Output:** Selected sample set $S$
1: Initialize $S \leftarrow \emptyset$
2: **for** $c = 1$ to $C$ **do**
3:     $X_c \leftarrow$ all samples with label $c$
4:     Randomly select one seed sample $x_0 \in X_c$ and initialize $S_c \leftarrow \{x_0\}$
5:     **while** $|S_c| < k$ **do**
6:         **for** each $x_i \in X_c \setminus S_c$ **do**
7:             get mean cosine similarity by equation 2
8:             compute $\text{Score}(x_i)$ by equation 1
9:         **end for**
10:       Add sample with highest score to $S_c$
11:     **end while**
12:     Add $S_c$ to $S$
13: **end for**
14: **return** $S$

---

## C  DATAMAP CONSTRUCTED VIA DATASET CARTOGRAPHY

Figure 8 presents the data map constructed using training dynamics on the AFHQ dataset. The horizontal axis denotes sample variability, measured as the variance of predicted confidence across epochs, reflecting the stability of the model's predictions during training. The vertical axis shows the average confidence, representing the model's overall certainty for each sample. Based on these two metrics, the data map reveals three characteristic regions. Samples in the top-left area are easy to learn, with high confidence and low variability, indicating that the model consistently predicts them correctly with high certainty. In contrast, hard-to-learn samples cluster in the bottom-left, showing both low confidence and low variability; these are often misclassified in a consistent manner, possibly due to label noise or inherent ambiguity. Ambiguous samples appear in the top-right region, characterized by moderate-to-high confidence but high variability, suggesting unstable predictions and proximity to decision boundaries.

In addition to the 2D distribution, histograms on the right visualize the marginal distributions of confidence, variability, and correctness. Most samples exhibit high average confidence, while variability tends to stay below 0.5, indicating generally stable training. The correctness histogram shows a bimodal pattern, with peaks near 0 and 1, reflecting a mixture of consistently correct and incorrect samples. This structure highlights the existence of both typical and boundary-challenging samples, offering valuable guidance for designing informed augmentation strategies.

## D  MAPPING FUNCTION ARCHITECTURE AND TRAINING DETAILS

**Architecture.** The mapping function $f : \mathcal{Z}_c \to \mathcal{Z}_s$ is implemented as a multilayer perceptron (MLP) with several fully connected layers, each followed by LayerNorm and ReLU activation to enhance nonlinear representation capability and stabilize gradient propagation. The network takes feature embeddings from the downstream classifier latent space $\mathcal{Z}_c$ as input and outputs vectors with the same dimensionality as the Stable Diffusion VAE latent space $\mathcal{Z}_s$.

**Training Strategy.** To ensure high-quality mapping, we propose a dual statistical alignment scheme. During training, we jointly minimize the mean squared error (MSE) between the mapped distribution's mean $\hat{\mu}_s$ and the reference mean $\mu_s$ of the Stable Diffusion VAE, as well as the MSE between the mapped log–variance $\log \hat{\sigma}_s^2$ and the reference $\log \sigma_s^2$:

$$L = \text{MSE}(\hat{\mu}_s, \mu_s) + \gamma \, \text{MSE}(\log \hat{\sigma}_s^2, \log \sigma_s^2), \tag{13}$$

where $\gamma$ balances the mean and variance constraints (commonly set to $\gamma = 0.5$). During optimization, both the classifier and the Stable Diffusion VAE are kept frozen, and only the parameters of the

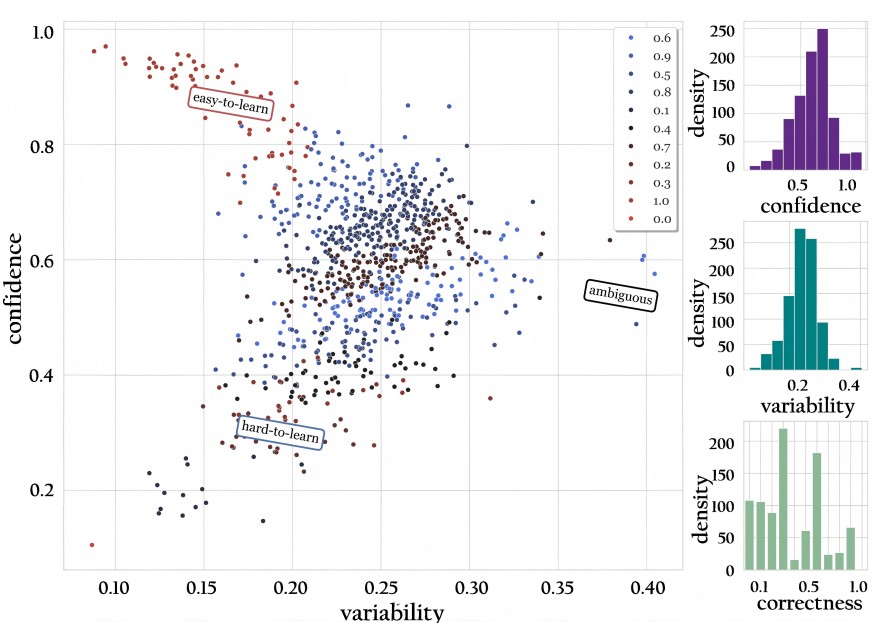

Figure 8: Training dynamics based data map on the AFHQ dataset with ResNet-50.

mapping network are updated to preserve the original semantic structures of the latent spaces. Training uses the Adam optimizer with an early stopping strategy to monitor validation loss. LayerNorm and GELU activations are applied in every MLP layer to suppress gradient explosion or vanishing, while weight decay is introduced to further improve generalization.

## E  EXPERIMENTAL DETAILS

**Augmented Sample Settings Across Different Methods.**   As shown in Table 3, our proposed augmentation strategies, ExploreAugment with BECS and Cartography, require substantially fewer additional samples than traditional methods. Specifically, the number of generated samples is only about $0.10{\sim}0.18\times$ of the original training set, while methods like Cutout, Random Erase, and YONA expand the dataset by $1.0\times$, effectively doubling the sample count. Despite using far fewer generated samples, our methods achieve comparable or even superior performance on multiple datasets.

For example, on the Birds dataset with 84,635 original samples, BECS generates only about 13,520 new samples (approximately 16% of the original size), while Cutout and Random Erase add 84,635 samples each. Similarly, on the Pets dataset with 5,177 original samples, our methods generate only around 515 to 618 samples, significantly fewer than the 5,177 additional samples from other methods. Despite this drastic reduction in augmented data volume, our methods consistently achieve comparable or even superior model performance, highlighting their higher sample efficiency.

This efficiency stems from the targeted nature of our augmentation strategy, which focuses explicitly on ambiguous and decision-critical regions in the latent space. By avoiding redundant or low-value transformations and instead generating a small number of highly informative samples, our approach makes better use of the model's capacity. These results underscore the practical advantage of sample-efficient augmentation, especially in scenarios where labeled data are limited.

**Model Architecture and Hyperparameters.**   This study employs two mainstream classifier architectures: *ResNet-50* and *DeiT*.

We use the classic ResNet-50 deep convolutional neural network with 3 input channels. The number of output classes varies by dataset (e.g., 525 classes for the Birds dataset). The model is trained for 200 epochs using the Adam optimizer with an initial learning rate of 0.0001 and a weight decay of 1e-4. The learning rate scheduler is ReduceLROnPlateau, which dynamically adjusts the learning

Table 3: Number of augmented and total training samples across datasets and methods, along with the augmentation ratio relative to the original training set.

| Method | Augmented | Total | Ratio | Method | Augmented | Total | Ratio |
|---|---|---|---|---|---|---|---|
| **AFHQ (Train: 14630)** | | | | **Flowers (Train: 6552)** | | | |
| only real data | – | 14630 | 0× | only real data | – | 6552 | 0× |
| cutout | 14630 | 29260 | 1× | cutout | 6552 | 13104 | 1× |
| random erase | 14630 | 29260 | 1× | random erase | 6552 | 13104 | 1× |
| text2img | 10000 | 24630 | 0.68× | text2img | 6120 | 12672 | 0.93× |
| YONA | 14630 | 29260 | 1× | YONA | 6552 | 13104 | 1× |
| diffmix | 10000 | 24630 | 0.68× | diffmix | 4000 | 10552 | 0.61× |
| ours (BECS) | 1752 | 16382 | 0.12× | ours (BECS) | 1048 | 7600 | 0.16× |
| ours (Cartography) | 2628 | 17258 | 0.18× | ours (Cartography) | 1048 | 7600 | 0.16× |
| **Birds (Train: 84635)** | | | | **Pets (Train: 5177)** | | | |
| only real data | – | 84635 | 0× | only real data | – | 5177 | 0× |
| cutout | 84635 | 169270 | 1× | cutout | 5177 | 10354 | 1× |
| random erase | 84635 | 169270 | 1× | random erase | 5177 | 10354 | 1× |
| text2img | 62080 | 146715 | 0.73× | text2img | 5123 | 10300 | 0.99× |
| YONA | 84635 | 169270 | 1× | YONA | 5177 | 10354 | 1× |
| diffmix | 50781 | 135416 | 0.60× | diffmix | 5177 | 10354 | 1× |
| ours (BECS) | 13520 | 98155 | 0.16× | ours (BECS) | 515 | 5692 | 0.10× |
| ours (Cartography) | 11830 | 96465 | 0.14× | ours (Cartography) | 618 | 5795 | 0.12× |

rate based on validation performance, with a decay factor of 0.6 and patience of 10 epochs. The batch size is set to 512. Early stopping is applied with a patience of 200 epochs. Pretrained weights can be optionally loaded.

Another setup uses the Vision Transformer variant DeiT Tiny, trained for 100 epochs with batch size 512, Adam optimizer, and similar learning rate and scheduling parameters. The number of output classes varies by dataset, and pretrained weights are loaded.Both models utilize the standard cross-entropy loss function.

**Data Augmentation and Auxiliary Modules.** The interpolation-based augmentation module **DiffMorpher** enables adaptive instance normalization (AdaIN) and rescheduling mechanisms, with an interpolation depth of 5 frames. The Cartography module is used for identifying critical samples, filtering approximately 10% of the training set based on the threshold closeness metric, enabling dynamic sample selection and analysis during training. In the BECS configuration, the boundary control parameter $\alpha$ is set to 0.7, which regulates the proximity of the generated samples to the classifier's decision boundary in latent space.

## F  COMPARISON WITH TRADITIONAL AUGMENTATION UNDER PERFORMANCE CEILING

This experiment aims to highlight the clear gap between ExploreAugment and traditional augmentation methods by verifying that common transformations quickly hit a performance ceiling even when the amount of generated data is greatly increased. We construct a small, class-balanced subset of the AFHQ dataset to simulate a low-data regime.

Four widely used augmentation methods, **Horizontal Flip (HFlip)**, **Random Crop**, **Rotation**, and **ColorJitter**, are compared with **ExploreAugment**. For each method, we vary the augmentation multiplier:

$$\text{AUG\_MULTIPLIERS} \in \{1, 2, 3, 4, 5\}, \tag{14}$$

where a multiplier $k$ indicates that the augmented training set contains $k$ times the number of original samples. All methods are trained with the same classifier backbone, optimizer, and training schedule to ensure a fair comparison.

As shown in Figure 9, traditional augmentations quickly plateau after a multiplier of 2–3, with top-1 accuracy gains stalling around 45–57% despite continued dataset expansion. For instance, even at a 5× multiplier, Horizontal Flip, Random Crop, and Rotation remain below 50.5%, and ColorJitter, though the strongest baseline, stops near 57.52%. In sharp contrast, ExploreAugment consistently

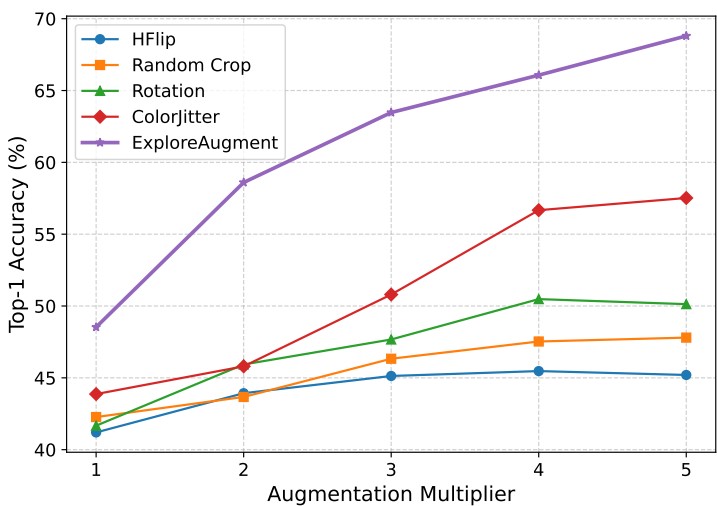

Figure 9: Performance Comparison of Different Augmentation Methods.

outperforms all baselines at every multiplier, starting from 48.53% at 1× and steadily climbing to 68.80% at 5×, showing continuous accuracy improvements where other methods saturate. In particular, ExploreAugment exceeds the best performance of all traditional methods even when generating fewer samples, highlighting its superior efficiency and effectiveness.

These results demonstrate that merely scaling up conventional transformations cannot overcome their inherent performance ceiling, whereas ExploreAugment leverages boundary-aware generation to achieve sustained performance gains with only a modest increase in computational cost.

THE USE OF LARGE LANGUAGE MODELS (LLMS)

LLMs were employed solely to aid or polish the writing of this paper, such as improving grammar, refining sentence structure, and enhancing clarity and readability, without altering the technical content.

ETHICS STATEMENT

Our research focuses on algorithmic methods for data augmentation and does not involve human subjects, personally identifiable information, or sensitive data. All datasets used are publicly available and have been widely adopted in prior research, with usage fully compliant with their respective licenses. No harmful insights, privacy risks, or security concerns are introduced by our methods. We have made every effort to ensure fairness, transparency, and research integrity throughout the study.

REPRODUCIBILITY STATEMENT

We have taken extensive steps to ensure the reproducibility of our work. All model architectures, training configurations, and hyperparameter settings are described in detail in Sections 4–5 of the main paper, with additional experimental settings provided in Appendix B–E. An anonymous implementation of our method, including training scripts and instructions, is linked in the main text, and code of the proposed algorithms is provided in the supplementary materials to facilitate replication of our results.

