# OpenReview forum: "ExploreAugment: Adaptive Exploratory Data Augmentation based on Boundary Awareness"
_ICLR.cc/2026/Conference — ICLR 2026 Conference Withdrawn Submission_

### Official Review · Reviewer_bhmS · 2025-10-25

**Soundness:** 3
**Presentation:** 3
**Contribution:** 3
**Rating:** 6
**Confidence:** 4

**Summary:**

This paper is focused on improving upon existing data augmentation techniques by augmenting the most informative data points using model-aware selection. The method leverages three stages: first it uses a boundary aware selection strategies to identify critical samples, then using boundary sample generation to generate both boundary-ambiguous and semantically coherent images, and finally using a fusion strategy to incorporate the augmented data points into the training loop.

**Strengths:**

The main strength of this paper is the limited number of augmented samples that are required to achieve similar levels of performance. This will significantly reduce the memory requirements of storing the augmented samples and the training time for image classification. Additionally, the boundary aware strategy is novel, and the paper is well-motivated and each part is well written.

**Weaknesses:**

There a few weaknesses for this paper.
1) First, there is no comparison to existing model-based / model-aware methods such as SoftAugment [1] or SAFLEX [2]. Such methods leverage the model's predictions to generate new samples, and would be good comparisons to the current approach to strengthen the paper. These should also be added to the related works section.
2) The boundary aware selection strategy seems rather heuristic. The authors should explain why this composite score makes sense and add ablation studies for each individual metric (entropy and consistency).

[1] Liu, Y., Yan, S., Leal-Taixé, L., Hays, J., & Ramanan, D. (2023). Soft augmentation for image classification. In Proceedings of the IEEE/CVF Conference on Computer Vision and Pattern Recognition (pp. 16241-16250).
[2] Ding, M., An, B., Xu, Y., Satheesh, A., & Huang, F. (2024). SAFLEX: Self-Adaptive Augmentation via Feature Label Extrapolation. arXiv preprint arXiv:2410.02512.

**Questions:**

1) The authors showed that with a small number of additional augmentations the performance matches existing augmentation methods. However, does this performance continue to increase as we add more augmentations (up to 60% augmentations instead of 10-20%)?
2) The method requires training a model to match the classifier latent space with the pretrained diffusion model latent space. What is the training time for this, and does this have to be retrained for different classifier / diffusion model pairs?
3) Building on Question 2, can the authors provide a efficiency comparison to other methods (wall clock time or FLOPs)? If the time it takes to generate 20% augmentations is more than the time it takes to generate 60% augmentations for other methods, this decreases the value of the method.
4) How necessary are the filtering mechanisms for the fusion training?

---

### Official Review · Reviewer_4foe · 2025-10-27

**Soundness:** 1
**Presentation:** 2
**Contribution:** 2
**Rating:** 2
**Confidence:** 4

**Summary:**

This paper proposes the ExploreAugment framework to address the inefficiency of traditional data augmentation, which is often decoupled from model training, by focusing on improving the coverage of decision-critical regions. It employs a model-aware, adaptive closed-loop system: first, it identifies uncertain "key samples" near the decision boundary using model feedback; second, it generates "boundary-ambiguous" new samples using diffusion models and LoRA interpolation (DiffMorpher); finally, it filters these samples via a two-stage filter and injects them using a progressive weighting scheme, enabling dynamic coordination between the classifier and augmentation.

**Strengths:**

1. Clear motivation.
2. The writing is overall good.
3. Good ayalytical experiments.

**Weaknesses:**

1. While the paper prominently highlights "sample efficiency," it largely overlooks the framework's substantial computational overhead. This overhead stems from its iterative, multi-stage process involving inference, LoRA fine-tuning, diffusion sampling, and gradient filtering.
2. The proposed method has not been evaluated on large-scale datasets, e..g., ImageNet-1k. The empirical validation is confined exclusively to fine-grained classification datasets, a scope that does not fully align with the paper's broader claims of improving generalization.
3. Without reporting the computational costs compared to vanilla training and other augmentation methods, the practical significance weakens.
4. The compared baselines are out of date. Many widely used and more recent augmentation methods (e.g., SelectAugment, EntAugment, FreeAugment, TrivialAugment...) are not compared. Thus, the claims made in this paper are not well validated.
Based on the weakness above and the questions, I vote for rejection at the current stage. I will consider updating my score based on the author's response.

**Questions:**

Could you please quantify the computational overhead of the ExploreAugment framework? While the paper emphasizes "sample efficiency," it omits a discussion of the "wall-clock time." Given the iterative, multi-stage process involving inference, LoRA fine-tuning, diffusion sampling, and gradient filtering, what is the approximate factor increase in total training time on the AFHQ dataset compared to a standard baseline like Cutout?
The 'boundary ambiguity' targeted in fine-grained tasks (e.g., distinguishing between species) differs significantly from the 'high inter-class variance' present in general-domain benchmarks like ImageNet. Could you provide further evidence or justification that your method remains effective for non-fine-grained tasks, especially given the absence of validation on such general-domain benchmarks?
In the Stage 3 auxiliary selection mechanism, the confidence interval [p_min, p_max] is used to filter generated samples. This appears to be a critical step, yet the paper lacks discussion on how these hyperparameters were set and provides no sensitivity analysis. Could you please specify the values used for p_min and p_max in your experiments, and elaborate on how sensitive the final model performance is to this choice?

---

### Official Review · Reviewer_2b1M · 2025-10-31

**Soundness:** 1
**Presentation:** 2
**Contribution:** 2
**Rating:** 2
**Confidence:** 5

**Summary:**

Traditional DA often applies uniform transformations across all samples, leading to redundant data expansion and inefficient training. This paper proposes ExploreAugment that dynamically targets and refines decision-critical regions in the latent space.
While there are some strengths, some critical weaknesses exist. Thus, at this stage, I vote for 2-reject, but I will consider updating my score based on the authors' responses.

**Strengths:**

- The proposed method is dynamic, adaptive, and model-aware, beyond static methods.
- The proposed method incorporates a generative model to enhance sample diversity.
- The paper contains many analytical experiments, which is good.

**Weaknesses:**

- Some recent methods also propose dynamic and adaptive DA mechanisms (e.g., EntAugment, FreeAugment, TeachAugment, AdaAugment, MADAug, etc.). However, none of these related works have been introduced in the related literature or experimental section. - The authors are strongly advised to discuss the difference between the proposed method and these related works.
- Regarding the main experiment, the compared methods are too limited. In fact, methods such as Cutout and RandomErase are weak baselines in data augmentation. More recent and stronger baselines are strongly suggested to compare with to validate the empirical performance of the proposed method.
- In line 13 of Algorithm 1, it seems that the augmented samples are incorporated into the original datasets. Thus, during comparison, the augmented data volume of different methods should be the same, i.e., Aug. Ratio in Table 1. The current comparison seems unfair.
- In Eq. (9), the author uses classifier confidence to select samples, while this may be unreliable.
- One of the most important evaluation metrics of DA is efficiency. One concern is that using diffusion-based generation during online training may introduce noticeable training costs, which will significantly reduce the effectiveness of DA methods. Without reporting the additional training costs of the proposed method, the effectiveness of the proposed method degrades.
- In the Abstract, the authors claim that the proposed method is model-aware, while in the Contributions,  they say that the method is model-independent. It requires further clarity.

**Questions:**

- Did the author try to train models only using the augmented data? This is widely used in most DA baselines.
- Did the author evaluate the performance of different generation methods on model training, not just visualization results, in Section 5.3 in Figure 5?
- The authors claim that the proposed method is boundary-aware. Does the augmented data utilize the same boundary?

---

### Official Review · Reviewer_f6ik · 2025-10-31

**Soundness:** 2
**Presentation:** 3
**Contribution:** 2
**Rating:** 4
**Confidence:** 4

**Summary:**

The paper proposes ExploreAugment, a model-aware, boundary-targeted augmentation framework. In the first stage, the method selects key training samples that are likely near the classifier’s decision boundary using either BECS or Dataset Cartography. In the second stage, it generates boundary-ambiguous but semantically valid images via diffusion-based latent interpolation (LoRA-adapted Stable Diffusion + SLERP over endpoints). In the third stage, it injects the curated synthetic samples back into training with a ramp-up weighting and a two-stage filter. Across AFHQ, Flowers, Birds-525, and Oxford Pets with ResNet-50 and DeiT backbones, ExploreAugment reports higher top-1 accuracy than baselines such as cutout, random-erase, Text2Img, YONA and DiffMix, while using only ~10%–20% extra samples.

**Strengths:**

1. The paper proposes a closed-loop framework for data augmentation that incorporates key sample identification via BECS or Cartography and sample generation via diffusion models. Generated images, under proper guidance, can potentially serve as hard samples for classification tasks.

2. The paper showed improved classification accuracies using less augmented data (10%-20% augmentation ratio), on various datasets and two representative model architectures.

**Weaknesses:**

1. The motivation of using diffusion models for data augmentation is unclear. While diffusion models have achieved significant success for image generation, the latent spaces for generative and discriminative models can be quite different. And generating similar-looking samples does not fully exploit the generation capacity of diffusion models.

2. The proposed method incurs significant computation overhead, as it needs to LoRA-update the diffusion model twice for each selected pair of samples from different classes. The method is hard to scale to large datasets, such as ImageNet or larger sets. While the method requires less augmented samples to reach competitive performance, the actual compute consumption can still be significant compared to other methods, which the paper does not discuss in details.

**Questions:**

1. What is the compute footprint relative to other methods such as cutout and Text2Img under equal accuracy?

2. The generated samples tend to have a drifted distribution with low contrast, as shown in Figure 1 and Figure 5(a). This can be due to capability limit of the diffusion model, insufficient sampling steps, information loss in latent space mapping, or other reasons. Is there any study for this phenomenon?

---

### Note · Authors · 2025-11-13

I have read and agree with the venue's withdrawal policy on behalf of myself and my co-authors.